# Factors Related to Physical Activity among Older Adults Who Relocated to a New Community after the Kumamoto Earthquake: A Study from the Viewpoint of Social Capital

**DOI:** 10.3390/ijerph20053995

**Published:** 2023-02-23

**Authors:** Yumie Kanamori, Ayako Ide-Okochi, Tomonori Samiso

**Affiliations:** 1Graduate School of Health Sciences, Kumamoto University, Kumamoto City 862-0976, Japan; 2Health and Welfare Policy Division, Health and Welfare Bureau, Kumamoto City 860-0808, Japan

**Keywords:** earthquake, relocation, physical inactivity, community participation, social support

## Abstract

Previous studies have shown an association between social capital and physical activity in older adults. Older adults who relocated after the Kumamoto earthquake may become physically inactive, and the extent of this inactivity may be buffered by social capital. Accordingly, this study applied the social capital perspective to examine factors that affect the physical activity of older adults who relocated to a new community after the Kumamoto earthquake. We conducted a self-administered mail questionnaire survey with 1494 (613 male, 881 female, mean age 75.12 ± 7.41 years) evacuees from temporary housing in Kumamoto City, aged 65 years and above, who relocated to a new community after the earthquake. We performed a binomial logistic regression to examine the factors affecting participants’ physical activity. The results showed that physical inactivity (decreased opportunities for physical activity, decreased walking speed, and no exercise habits) was significantly associated with non-participation in community activities, lack of information about community activities, and being aged 75 years and over. Lack of social support from friends was significantly associated with lack of exercise habits. These findings encourage participation in community activities, alongside giving and receiving social support in health activities that target older adults who relocated to new communities after the earthquake.

## 1. Introduction

The Kumamoto earthquake that occurred in April 2016 registered an intensity value of 6 on the Richter scale throughout Kumamoto City [1]. Located approximately 13 kms from the epicentre, Kumamoto City experienced two earthquakes that resulted in approximately 136,000 damaged dwellings, and up to 11,000 people were forced to relocate to temporary housing [1]. Around 2400 (21%) of the individuals who relocated were older adults. In a survey conducted among those who left the temporary housing four years after the disaster, about 34% had moved from the community where they had lived before the disaster to another community [2].

A survey by Hikichi et al. found that individual relocation was associated with a decline in the daily living activities of disaster victims 2.5–5.5 years after the Great East Japan Earthquake [3]. Tsuji et al. also point out that depressive symptoms increased after the Great East Japan Earthquake [4] and that depression in older adults means that they stay at home [5], which may be a factor in their physical inactivity after the earthquake. They then state that walking and participation in group exercise are necessary to prevent depression in older adults after the earthquake [6]. Furthermore, physical activity is necessary to prevent the development of frailty [7]. However, Murakami et al. found that the physical activity level of evacuees living in temporary housing in Kamaishi, Iwate Prefecture, was lower than that of the national average [8]. Furthermore, Ito et al. showed that a significantly higher percentage of older adults who lived in different rental housing than before the Great East Japan Earthquake were less physically active than those living in the same housing as before the earthquake [9]. In addition, a study comparing pre- and post-earthquake physical activity found that in areas with strong social connections, the decline in daily physical performance after the disaster was about one third [10]. Considering these findings, it can be assumed that older adults who relocated to another community after the earthquake are more likely to be inactive. However, social connections, i.e., social capital, may prevent physical inactivity.

The public health field considers social capital one of the most important factors that influence people’s health and sense of wellness [11]. Especially in public health crisis situations such as natural disasters, it has been demonstrated that the richer the social capital, the more that health damage can be controlled and the faster the recovery [12]. Therefore, we reviewed the previous studies on the relationship between social capital and physical activity, such as those conducted in Korea, Finland, Japan, and other countries [13,14]. Specifically, in a study of Korean adults, physical activity levels, defined by intensity, duration, and frequency, were higher when there was participation in community activities [13]. A study of 8000 adults in Finland also found that those who participated in community activities were more physically active [14]. Furthermore, a study conducted in Japan found that older adults with high social activity levels were more physically active and had shorter passive sitting times [15].

According to Kawachi et al., forms of social capital, such as participation in community organisations and social support, are directly related to health outcomes [16]. Therefore, we considered the relationship between social support and physical activity. In a study by Goodwin et al. that tracked psychological distress after the Great East Japan Earthquake, those with low physical activity levels and no social support after the disaster had lower levels of psychological health [17]. Moreover, Teramoto et al. found that after the Great East Japan Earthquake, social relationships with friends were associated with reduced psychological distress, especially among females aged 65 years and over [18]. Loss of social contact impacts psychological health, which, in turn, decreases physical function [19]. In a study by Sato et al. after the Kumamoto earthquake, the strength of social cohesion contributed to the reduction in depression among women [20]. A previous Kumamoto Prefecture reconstruction survey showed that 43.4% of the population had decreased their opportunities for physical activity after relocating due to the earthquake [21], a situation in which physical activity should be promoted through social capital.

Social support refers to support sought from others to cope with everyday problems [16]. Therefore, social support, defined as having friends and peers with whom one can discuss daily problems, is thought to maintain social contact and psychological health and prevent physical inactivity. Among other things, connections with friends [22], neighbours [23] and district welfare commissioners affect mental health issues [24] such as loneliness, as well as quality of life measures, such as happiness. Furthermore, in a study conducted in Miyagi Prefecture on residents who migrated after the Great East Japan Earthquake, the social capital of individual relocation was lower than that of community-based group relocation [3]. Based on this finding, it is thought that the individuals who relocated may have experienced difficulty in fostering social capital in their new communities after the earthquake, which may have affected their physical activity, especially among older adults.

In Kumamoto City, there was no group relocation by community units after the earthquake. Therefore, it is possible that physical inactivity may be related to social capital among older adults who relocated to a new community after the earthquake in Kumamoto City. However, we did not find any clear findings on this issue in the previous research. Therefore, we examine whether social capital can explain physical inactivity among older adults who relocated to a new community in Kumamoto City after the Kumamoto earthquake. We explore the factors that affect physical activity among the older adults who relocated from the perspective of social capital so as to provide suggestions for future health activities.

## 2. Materials and Methods

### 2.1. Participants

This study followed a cross-sectional design. The sample comprised of 2016 Kumamoto earthquake victims who had moved out of temporary housing by December 2019, were aged 65 years and over, resided in Kumamoto City, and had relocated to a different community after the earthquake. A self-administered questionnaire was mailed to 11,479 households who had relocated to be completed by individuals aged 18 years and over in each household. Completed questionnaires were returned. Of the individuals who responded, those aged 65 years and over who had relocated to another community were included in the study. Of the 8966 collected questionnaires, 1494 persons (41.0% male, 59.0% female) were selected for analysis; these included adults aged 65 years and over who had relocated to a new community after the earthquake (Figure 1). The data were collected from July to December 2020. In Kumamoto Prefecture, several local authorities have conducted recovery surveys. The recovery survey conducted by Kumamoto Prefecture had already shown that changes in the community after the earthquake reduced opportunities for physical activity. The recovery survey had also been used for research purposes and had shown, for example, mental health. However, this was the first recovery survey to incorporate pieces for older adults. Through this survey, Kumamoto City has ascertained the health status of its citizens. Through this survey, the need for future research and support will be considered [25,26]. The study was reviewed and approved by the Institutional Review Board of Kumamoto University (approval no. 1940, approved on 4 June 2020), Kumamoto, Japan.

In this study, we defined ‘community’ as a social group determined by geographical boundaries, such as administrative areas [27]. We defined ‘physical activity’ as all movements performed in daily life, such as walking, housework, daily activities, and exercise [28]. We define social capital as ‘social connections and the norms and trust that emerge from them, characteristics of social organizations that effectively lead to coordinated action’ [29]. ‘Older adults’ shall be defined as respondents aged 65 years and over. An ‘Exercise habit’ is defined as performing household chores that replace exercise or physical activity at least once a week.

### 2.2. Variables

#### 2.2.1. Physical Activity

To understand the changes in opportunities for physical activity before and after the earthquake, we asked the participants, ‘To what extent have the opportunities for physical activity in your daily life changed compared to before the Kumamoto earthquake? Please check all that apply’. The response options were (1) large increase, (2) slight increase, (3) unchanged (active), (4) unchanged (inactive), (5) slight decrease, and (6) large decrease. The questions were set up based on a previous recovery survey conducted by Kumamoto Prefecture, which found that respondents who had a change of address after the earthquake [21] were less likely to be physically active. We also asked, ‘Do you think your walking speed has become slower compared to before? Please check all that apply’. The response options were (1) yes and (2) no. Walking speed was based on the standard questionnaire items during health checks [30] and the Okazaki et al. survey on perceptions of walking speed [31]. To evaluate the participants’ exercise habits, we asked, ‘Do you walk or do household chores (cleaning, gardening, etc.) that constitute exercise at least once per week?’ The response options were (1) yes and (2) no. The questions on exercise habits were set up with reference to the Active Guide [32] based on the Physical Activity Reference for Health Promotion 2013.

#### 2.2.2. Attributes

We collected information on participants’ basic attributes, such as sex, age, cohabitants, temporary housing category, and current residence. Age was recorded in a descriptive form by asking the participants to write their current age. Sex was recorded using the Kumamoto City ledger. To assess whether the participants had cohabitants, we asked, ‘Do you have anyone living with you?’ The response options were (1) yes and (2) no. The classification of temporary housing was recorded using the Kumamoto City ledger. We classified the participants’ residence type using the following response options: (1) owned house, (2) houses for rent, (3) public housing, (4) public housing for disasters, (5) hospital and institutions, and (6) other.

#### 2.2.3. Social Capital

We classified social capital in terms of participation in community activities and the availability of social support (advisory partners). The types of consulting partners were friends, neighbours, and district welfare commissioners. Participation in community activities was based on the question, ‘Do you participate in events and social gatherings held in your community?’ The response options were (1) I participate, (2) I do not participate, or (3) I do not know about such information (non-information). The question was set based on the fact that the same question had been asked in a previous Kumamoto Prefecture recovery survey; non-participation was found to be 60.2%, and non-information was found to be 8.2% [21]. Additionally, in the study by Hikichi et al. which revealed social capital and mental and physical health, the question “Do you participate in any local events?” also referred to this question [3]. To determine participants’ social support, we asked, ‘Who do you consult with about your problems?’ The response options were (1) family; (2) friends; (3) neighbours; (4) co-workers; (5) district welfare commissioners; (6) medical institutions; (7) welfare offices, such as nursing care facilities; (8) city hall/ward office; and (9) no one. As social support, the role of friends [22], neighbours [23] and district welfare commissioners [24] is particularly important. In addition, medical institutions and care providers are involved according to health status [33]. 

### 2.3. Data Analysis

After calculating the basic statistics, we performed a Chi-square test of independence (Χ^2^ test) to examine the relationship between physical activity, basic attributes, and the social capital influence variables. Next, we conducted a binomial logistic regression analysis to examine the factors that influenced physical activity. Using decreased opportunities for physical activity after the earthquake as the dependent variable, we examined whether decreased walking speed and lack of exercise habits were applicable.

We created three models for this study (Figure 2). Model I explains how physical activity is affected by the presence or absence of participation in community activities. Model II explains how physical activity is affected by the addition of basic attributes to Model I. Finally, Model III explains the impact on physical activity by adding social support to Model II. Model I utilises a univariate analysis, while Model II and Model III utilise multivariate analyses.

The independent variables were sex, age (whether the person falls into the category of 75 years and over), presence or absence of a cohabitant, temporary housing category, current residence, participation in community activities, and presence or absence of social support (friends, neighbours, or district welfare commissioners). We created dummy variables for the independent variables. Before conducting the logistic regression analysis, we calculated the Spearman’s rank correlation coefficient to confirm multicollinearity and ensure that the correlation coefficient did not exceed 0.8. We selected the variables using the forced entry method and set the statistical significance level as 0.5% (two-sided). We conducted Χ^2^ and Hosmer-Lemeshow (HL) tests on the models to assess model fit. To assess model fit, we calculated the Nagelkerke R^2^ values and excluded the missing values for each variable. We used SPSS Statistics 27.0 for Windows as the statistical software.

## 3. Results

There were 613 (41.0%) male and 881 (59.0%) female respondents. The average age of the participants was 75.12 ± 7.41 years (65–105 years); 807 (54.0%) were aged 65–74 years and 687 (46.0%) were aged 75 years and over. The basic attributes of the respondents analysed are shown in Table 1. The participants’ demographics of the analysed participants are shown in Table 1. Table 1 shows that there are 613 (41.0%) males and 881 (59.0%) females. The mean age is 75.12 ± 7.41 years (65–105). Of the total participants, 807 (54.0%) are aged 65–74 years, and 687 (46.0%) are aged 75 years and over. Furthermore, 899 (60.2%) live with a cohabitant, while the largest number of participants [1255 (84.0%)] live in temporary housing in the private sector. At the time of the survey, 209 (14.0%) participants were living in an owned house, 592 (39.6%) were living in a house for rent, and 568 (38.0%) were living in public housing. The results show that 300 (20.1%) participate in community activities, while 960 (64.3%) do not participate in community activities, and 187 (12.5%) have no information about such activities. Regarding social support, 497 (34.2%), 83 (5.7%), and 34 (2.3%) of the participants consult friends, neighbours, and district welfare commissioners, respectively, when they have problems.

Table 2 shows the participants’ physical activity after relocation due to the earthquake. Regarding the changes in opportunities for physical activity after the earthquake, 403 (27.0%) participants state that their opportunities have considerably decreased, while 322 (21.6%) state that their opportunities have somewhat decreased. Moreover, 983 (65.8%) state that their walking speed has decreased and 995 (66.6%) report having exercise habits.

Table 3 shows the results for the cross tabulation of physical activity and the independent variables. Age, type of temporary housing, and social support from friends are significantly associated with decreased opportunities for physical activity after the earthquake. Age, participation in community activities, and social support from district welfare commissioners are significantly associated with decreased walking speed after the earthquake. Sex, age, current residence, participation in community activities, and social support from friends and from neighbours are significantly associated with the presence of exercise habits.

Table 4 shows the results for the binomial logistic regression analysis, with physical activity as the dependent variable and participation in community activities, basic attributes, and social support as the independent variables. Regarding the Χ^2^ test results for the models, Model I shows that *p* = 0.121 for decreased opportunities for physical activity after the earthquake. All other results are significant, and the HL test results are *p* ≥ 0.05. We further checked the Nagelkerke R^2^ values to select the model with the best fit in each dependent variable. The results show that Model III has the highest values for all three aspects of physical activity, at 0.063 for decreased opportunities for physical activity, 0.118 for decreased walking speed, and 0.110 for no exercise habits. Therefore, from the binomial logistic regression analysis results, we adopted the values of Model III.

Non-participation in community activities (1.38, 1.04–1.83), being unaware of such information (1.71, 1.15–2.56), and being aged 75 years and over (2.11, 1.69–2.65) are more likely to decrease opportunities for physical activity after the earthquake. Non-participation in community activities (1.67, 1.23–2.27), non-information about such activities (3.59, 2.22–5.80), and being aged 75 years and over (3.01, 2.31–3.92) are more likely to be associated with decreased walking speed. Lack of exercise habit is associated with non-participation in community activities (2.42, 1.69–3.47), non-information about such activities (3.05, 1.91–4.86), being male (2.14, 1.67–2.74), being aged 75 years and over (1.55, 1.20–1.99), and having no social support/friends (1.44, 1.09–1.91); thus, participants who fall under these categories are less likely to maintain exercise habits. Conversely, participants living in a house they owned (0.29, 0.12–0.73), houses for rent (0.32, 0.14–0.77), and in public housing (0.34, 0.14–0.82) are less likely to be included in the no exercise habits category.

## 4. Discussion

### 4.1. Participants’ Physical Activity

In 2020, the population of Kumamoto City totalled 738,567 people [34], of which 110,750 people were evacuees (Kumamoto Earthquake Kumamoto City Earthquake Record Magazine, 2016) and approximately 24,000 people were aged 65 years and over. Therefore, the 1494 participants whose data were analysed herein correspond to 6.4% of the total population. A previous study found that 5.3% of the total population had relocated to temporary housing [3]; our study observed a similar trend.

Overall, we found that 48.6% of the participants fell into the category of decreased opportunities for physical activity. In 2020, Kumamoto Prefecture surveyed the health of those affected by the Kumamoto earthquake who were living in temporary housing in 17 municipalities; 32.8% had decreased opportunities for physical activity [21], and the percentage of decliners was lower than 48.6% in this study. In a study of 81 older adults who relocated from their homes to senior housing in Finland, the average walking speed per second decreased by 0.11 s after relocation [35]. However, Okazaki et al. surveyed lifestyle factors related to malnutrition in Fukushima Prefecture after the Great East Japan Earthquake, and found that 55.6% of the participants perceived that their walking speed was not fast enough [31]. Meanwhile, our study found that 65.8% of participants felt that their walking speed had decreased. A previous study found that 37.6% (41.9% of males and 33.9% of females) of older adults in Japan had exercise habits [36]. Meanwhile, our study found that 66.6% of the participants had exercise habits, which is relatively higher than the general ownership rate. In the aforementioned study by Okazaki et al. in Fukushima Prefecture, 64.6% of the participants had exercise habits [31]. This value is comparable to that of our study. Although Fukushima Prefecture was affected by the Fukushima Daiichi Nuclear Power Plant accident, we used it as a reference to compare the exercise habit rate of older adults living in the area after the earthquake. The results suggest that older adults may have a higher exercise habit rate compared to the general population after the earthquake. Based on the above, we consider our study participants a population that is as inactive as in the previous studies in terms of decreased opportunities for physical activity and decreased walking speed; moreover, their exercise habits are relatively similar to those of older adults after the earthquake.

### 4.2. Factors Associated with Physical Inactivity

We found that social capital was significantly associated with physical inactivity among older adults who relocated to a different community from that of their pre-earthquake community after the Kumamoto earthquake. The positive association between participation in community activities and physical activity has been shown in Kim et al. and Nieminen et al. [13,14]. In Korean adults, higher levels of social participation and generalised trust have been found to significantly increase levels of physical activity [13]. In Finnish adults, more social participation and networking have been found to significantly increase physical activity in leisure time [14]. Furthermore, low social capital at the state and county level has been found to be significantly associated with physical inactivity in a multilevel analysis conducted in the United States [37]. In addition, A study after the Armenian earthquake showed that older age significantly decreased physical activity, producing 1.07 times more mobility difficulties and 1.05 times worse usual activity. On the other hand, higher social support scores decreased mobility difficulties and usual activity worse by 0.91 times, indicating the effect of social capital on physical activity [38]. Our results also showed that those who did not participate in community activities had decreased opportunities for physical activity, which was consistent with the results of the previous studies. However, to the best of our knowledge, our study is the first to identify an association between social capital and physical activity in older adults who relocated after the Kumamoto earthquake. Social capital can be reduced through unavoidable migration after a disaster [39]. Moreover, the migration of older adults can result in adaptation problems to their new community, affecting their physical and mental health [40]. Therefore, the decrease in opportunities to go outside after the earthquake [21] may be a trigger for physical frailty. Our study provides valuable insight into the prevention of physical inactivity among earthquake victims.

Simultaneously, we found that lack of information about community activities was associated with more physical inactivity than non-participation. In Kumamoto City, information about community activities is mainly communicated through posts, circulars, verbal communication at meetings, and invitations among residents. Elena et al. indicate that health information may motivate individual health behaviours by turning to it for health-related purposes [41]. Therefore, information about community activities may not catch the attention of those who have no awareness of health issues. Furthermore, there is a risk that the information disseminated to members within a current circle of residents may not reach new residents. Therefore, for the older adults who relocated to a new community after the earthquake, the necessary information should be delivered by professionals and key persons in the community.

Regarding sex, we found that among the older adults who relocated after the earthquake, males showed a significantly higher association with lack of exercise habits than females. To date, there is no study that has identified exercise habits by sex among older adults who relocated to a new community after the earthquake. However, it is noteworthy that even in normal times, males are less likely to maintain exercise habits than females [42], which is consistent with our study results. Thus, in general, males tend to have fewer exercise habits than females. Therefore, we believe that health activities that particularly take males into consideration are required.

Regarding age, our results found that those aged 75 years and over were significantly more likely to have decreased opportunities for physical activity, decreased walking speed, and no exercise habits than those aged 74 years and under. In a study on orthopaedic diseases among the victims of the Great East Japan Earthquake, the prevalence rate for back and joint pain in the limbs was relatively high among those aged 75 years and over [43]. Muscle weakness inevitably occurs with age [44].

Furthermore, as mentioned earlier, preventive care activities in Kumamoto City are conducted on a community basis; hence, older adults who relocated after the earthquake in this study adapted to their new community. Therefore, smooth relationships in a smooth new community make it easier to carry out health promotion activities and increase physical activity [45]. On the other hand, inability to adapt to the community may lead to physical inactivity. The results of this study showed that non-participants in community activities were more physically inactive than participants. This may have led to an inability to adapt to the new community, which may have affected physical inactivity. In addition, the self-restraint on opportunities for people to assemble due to COVID-19 may have also contributed to the difficulty in adapting to the community and the physical inactivity.

Finally, we found that social support from friends was associated with physical activity. We found that the absence of social support from friends was significantly associated with no exercise habits compared to the presence of social support from friends among older adults who relocated after the earthquake. Similarly, a study on middle-aged women in Chicago analysed the correlation between the physical activity of individuals and their friends; those who were continuously active had active friends [46]. Moreover, a systematic review of physical activity among older adults found that unaccompanied companionship was a disincentive for continuous physical activity, including exercise, suggesting the need for companionship [47]. Social support from friends is also believed to contribute towards the protection of psychological health [15,16]. Social cognitive theory states that an important driver of sustained behaviour is high self-efficacy [48]. Therefore, the presence of friends may increase sustained exercise, even after the earthquake. However, self-efficacy has been associated with the maintenance of physical activity for up to 12 months [49], and it is likely that we found a significant association with exercise habits in our study because of this. Therefore, when considering health activities that target older adults who relocated after the earthquake, it is necessary to focus on social support from friends as well. In view of the above, the ideal way to engage in habitual activities such as exercise is not only to participate in community activities, but also to develop friendships with other participants. Therefore, health workers need to pay attention to the frequency of activities and developmental stage of the new conurbations, and to create a familiar environment for older adults who have relocated after the earthquake.

### 4.3. Limitations and Significance

This study has the following limitations. First, this study was limited to the Kumamoto earthquake victims living in Kumamoto City; therefore, the results are not generalisable to all earthquake victims. Second, this was a cross-sectional study that was conducted four years after the Kumamoto earthquake. The survey was conducted when COVID-19 was first beginning to spread, so it is undeniable that the results may have been influenced by social behaviour restrictions. The period from July to December 2020, when the survey was conducted, was the period when measures to prevent the spread of COVID-19 were ordered. Participates were asked to refrain from assembly, although individual activities such as shopping were possible. It is therefore possible that the loss of opportunities for gathering and activity may have caused a low level of physical activity. To differentiate the effects of migration from those of COVID-19, a retrospective longitudinal study is necessary. Longitudinal studies can also confirm that causal relationships between variables cannot be determined. Furthermore, it is possible that there are independent variables that affect physical activity other than those included in our study models. Our survey did not ask if the earthquake caused locomotion, mental or other health problems. However, as previously mentioned, these could be factors influencing physical activity. In the future, it is also necessary to understand the actual situation of older adults with reduced physical activity and the means of activity they are seeking. Under COVID-19, digital media and use of the internet to connect was recommended, but older adults in Japan are not good at communicating using digital devices [50]. Additionally, as this study did not survey the social use of digital media, this needs to be studied in the future. Despite the limitations of our study, it is significant that we were able to clarify a previously unknown finding regarding the physical activity of older adults who relocated to a different community after the earthquake than before the disaster. We also believe that we have identified basic knowledge that is necessary for conducting health activities that target these older adults.

## 5. Conclusions

This study showed that social capital was significantly associated with physical inactivity among older adults who relocated to a different community from their pre- community after the Kumamoto earthquake. Physical inactivity (decreased opportunities for physical activity, decreased walking speed, and lack of exercise habits) was significantly associated with non-participation in community activities, non-information about such activities, and being aged 75 years and over. No social support from friends was also significantly associated with no exercise habits. The study suggests the need to foster social capital among older people who have moved to a new community. Health activities that target older adults who have relocated to a new community after the earthquake should take into account their participation in community activities as well as social support from friends.

## Figures and Tables

**Figure 1 ijerph-20-03995-f001:**
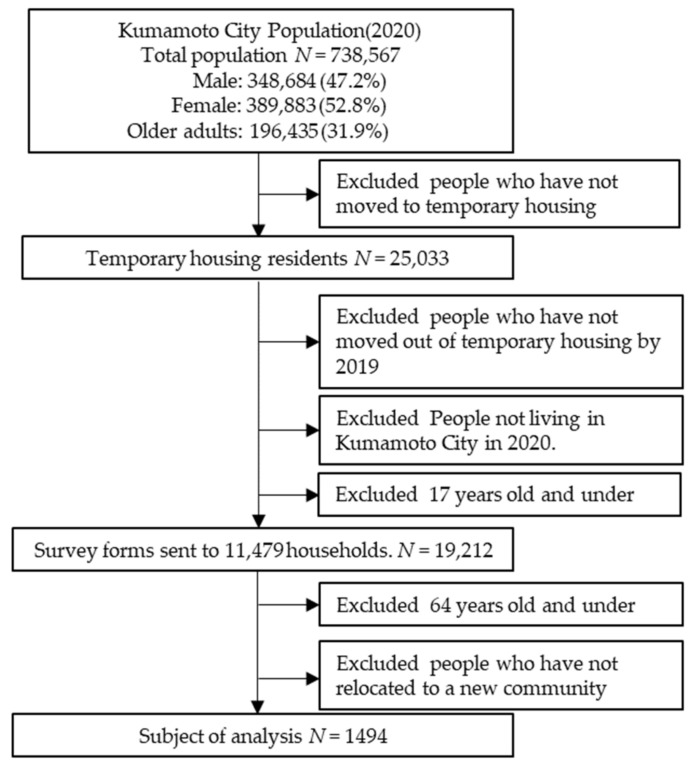
Population and data collection.

**Figure 2 ijerph-20-03995-f002:**
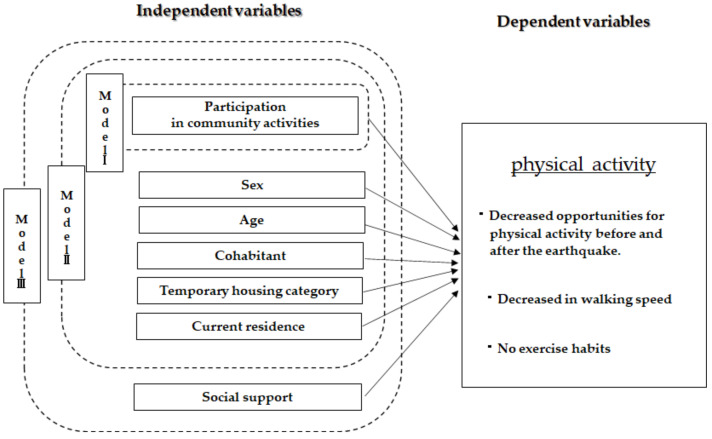
Model of analysis.

**Table 1 ijerph-20-03995-t001:** Participants’ demographics.

		*N* = 1494
	*n*	%
Sex		
Male	613	41.0
Female	881	59.0
Age		
Mean ± SD	1494	75.12 ± 7.41
65–74 years	807	54.0
75 years and over	687	46.0
Cohabitant		
Yes	899	60.2
No	584	39.1
Temporary housing category		
Prefabricated Temporary housing	73	4.9
Temporary housing in the private sector	1255	84.0
Temporary housing in the public sector	166	11.1
Current residence		
Owned house	209	14.0
Houses for rent	592	39.6
Public housing	568	38.0
Public housing for disaster	48	3.2
Hospitals and institutions	36	2.4
Other	33	2.2
Participation in community activities		
Yes	300	20.1
Non-participation	960	64.3
Non-information	187	12.5
Social support (friends)		
Yes	497	33.3
None	955	63.9
Social support (neighbours)		
Yes	83	5.6
None	1369	91.6
Social support (district welfare commissioners)		
Yes	34	2.3
None	1418	94.9

**Table 2 ijerph-20-03995-t002:** Prevalence rates of physical activity.

		*N* = 1494
	*n*	%
Opportunities for physical activity		
Large increase	43	2.9
Slight increase	117	7.8
Unchanged (active)	298	19.9
Unchanged (inactive)	262	17.5
Slight decrease	322	21.6
Large decrease	403	27.0
Decreased walking speed		
Yes	983	65.8
No	450	30.1
Exercise habits		
Yes	995	66.6
No	432	28.9

**Table 3 ijerph-20-03995-t003:** Cross tabulation of social support and independent variables.

											*N* = 1494				
	Decreased Opportunities for Physical Activity after the Earthquake					Decreased Walking Speed					No Exercise Habits				
	Applicable *n* = 725		Not Applicable *n* = 720		*p*-Value	Applicable *n* = 983		Not Applicable *n* = 450		*p*-Value	Applicable *n* = 432		Not Applicable *n* = 995		*p*-Value
	*n*	%	*n*	%		*n*	%	*n*	%		*n*	%	*n*	%	
Sex					0.165					0.563					<0.001
Male	284	39.2	308	42.8		397	40.4	189	42.0		228	52.8	358	36.0	
Female	441	60.8	412	57.2		586	59.6	261	58.0		204	47.2	637	64.0	
Age					<0.001					<0.001					0.001
65–74 years	328	45.2	456	63.3		465	47.3	320	71.1		208	48.1	572	57.5	
Over 75 years	397	54.8	264	36.7		518	52.7	130	28.9		224	51.9	423	42.5	
Cohabitant					0.746					0.642					0.860
Yes	439	61.1	433	60.2		590	60.3	264	58.9		254	59.5	596	60.0	
No	279	38.9	286	39.8		388	39.7	184	41.1		173	40.5	397	40.0	
Temporary housing category					0.047					0.164					0.320
Prefabricated temporary housing	45	6.2	27	3.8		44	4.5	21	4.7		20	4.6	46	4.6	
Temporary housing in the private sector	594	81.9	620	86.1		818	83.2	389	86.4		355	82.2	845	84.9	
Temporary housing in the public sector	86	11.9	73	10.1		121	12.3	40	8.9		57	13.2	104	10.5	
Current residence					0.128					0.158					0.042
Owned house	103	14.3	100	14.0		138	14.1	64	14.3		54	12.6	148	14.9	
Houses for rent	276	38.2	300	42.0		379	38.7	191	42.8		171	39.9	395	39.9	
Public housing	276	38.2	272	38.0		390	39.8	162	36.3		163	38.0	388	39.2	
Public housing for disaster	31	4.3	14	2.0		26	2.7	17	3.8		13	3.0	30	3.0	
Hospitals and institutions	20	2.8	14	2.0		21	2.1	8	1.8		16	3.7	13	1.3	
Other	16	2.2	15	2.1		25	2.6	4	0.9		12	2.8	17	1.7	
Participation in community activities					0.122					<0.001					<0.001
Yes	131	18.6	161	22.8		175	18.1	113	26.3		51	12.1	237	24.4	
Non-participation	476	67.5	458	65.0		646	66.9	282	65.6		305	72.1	621	64.0	
Non-information	98	13.9	86	12.2		145	15.0	35	8.1		67	15.8	113	11.6	
Social support (friend)					0.044					0.114					<0.001
None	480	68.0	445	62.9		647	67.0	271	62.6		314	74.8	600	61.6	
Yes	226	32.0	263	37.1		319	33.0	162	37.4		106	25.2	374	38.4	
Social support (neighbours)					0.365					0.065					0.004
None	661	93.6	671	94.8		917	94.9	400	92.4		407	96.9	906	93.0	
Yes	45	6.4	37	5.2		49	5.1	33	7.6		13	3.1	68	7.0	
Social support (district welfare commissioners)					0.290					0.021					0.565
None	687	97.3	695	98.2		937	97.0	429	99.1		409	97.4	953	97.8	
Yes	19	2.7	13	1.8		29	3.0	4	0.9		11	2.6	21	2.2	

Note: Pearson’s chi-square test. Fisher’s exact test. ‘Not applicable’ responses for ‘Decreased opportunities for physical activity after the earthquake’ include ‘significant increase’, ‘slight increase’, ‘unchanged (active)’, and ‘unchanged (inactive)’.

**Table 4 ijerph-20-03995-t004:** Association between physical activity and each independent variable.

													*N* = 1494					
	Decreased Opportunities for Physical Activity after the Earthquake						Decreased Walking Speed						No Exercise Habits					
	Model I		Model II		Model III		Model I		Model II		Model III		Model I		Model II		Model III	
	OR	95%CI	OR	95%CI	OR	95%CI	OR	95%CI	OR	95%CI	OR	95%CI	OR	95%CI	OR	95%CI	OR	95%CI
Participation in community activities (ref: yes)																		
Non-participation	1.28	0.98–1.66	1.36	1.03–1.80	1.38	1.04–1.83	1.48	1.12–1.95	1.73	1.28–2.33	1.67	1.23–2.27	2.28	1.64–3.18	2.39	1.69–3.39	2.42	1.69–3.47
Non-information	1.40	0.97–2.03	1.71	1.15–2.53	1.71	1.15–2.56	2.68	1.73–4.15	3.72	2.32–5.97	3.59	2.22–5.80	2.76	1.80–4.23	3.17	2.01–4.99	3.05	1.91–4.86
Sex (ref: female)																		
Male			0.91	0.73–1.13	0.91	0.72–1.13			1.05	0.82–1.34	1.06	0.82–1.36			2.19	1.72–2.79	2.14	1.67–2.74
Age (ref: 65–74 years)																		
Over 75 years			2.14	1.72–2.67	2.11	1.69–2.65			2.99	2.31–3.87	3.01	2.31–3.92			1.57	1.23–2.01	1.55	1.20–1.99
Cohabitant (ref: yes)																		
No			0.89	0.71–1.12	0.92	0.73–1.17			0.88	0.68–1.13	0.86	0.66–1.12			1.01	0.79–1.30	1.07	0.83–1.39
Temporary housing category (ref: public sector)																		
Prefabricated			1.4	0.75–2.64	1.36	0.71–2.57			0.93	0.45–1.91	0.92	0.44–1.90			0.75	0.37–1.52	0.81	0.40–1.66
Private sector			0.88	0.60–1.27	0.90	0.62–1.31			0.73	0.47–1.13	0.76	0.49–1.19			0.74	0.50–1.11	0.79	0.53–1.19
Current residence (ref: hospitals and institutions)																		
Owned house			1.33	0.58–3.03	1.51	0.65–3.55			1.14	0.41–3.19	1.18	0.42–3.32			0.33	0.13–0.80	0.29	0.12–0.73
Houses for rent			1.10	0.50–2.41	1.26	0.56–2.83			0.90	0.33–2.41	0.91	0.34–2.47			0.37	0.16–0.86	0.32	0.14–0.77
Public housing			1.18	0.54–2.58	1.34	0.59–3.02			1.19	0.44–3.20	1.26	0.46–3.41			0.39	0.17–0.92	0.34	0.14–0.82
Public housing for disaster			2.28	0.81–6.43	2.66	0.92–7.69			0.65	0.20–2.15	0.67	0.20–2.23			0.39	0.13–1.18	0.33	0.11–1.02
Other			1.13	0.39–3.28	1.31	0.44–3.88			2.46	0.58–10.52	2.52	0.59–10.8			0.59	0.19–1.83	0.52	0.17–1.62
Social support/friend (ref: yes)																		
None					1.16	0.91–1.48					0.96	0.74–1.25					1.44	1.09–1.91
Social support/neighbours (ref: yes)																		
None					0.75	0.46–1.22					1.59	0.95–2.65					1.56	0.82–2.98
Social support/district welfare commissioners (ref: yes)																		
None					0.76	0.35–1.66					0.34	0.12–1.02					0.61	0.27–1.38
Nagelkerke R^2^	0.004		0.060		0.063		0.021		0.111		0.118		0.031		0.093		0.110	

Note: We performed a binary logistic regression analysis. Variable selection was forced entry. *p* ≥ 0.05 for HL test results for all logistic regression analyses. The dependent variables were set at 1 (yes, decreased in opportunities for physical activity before and after the earthquake; decreased walking speed; and no exercise habits) and 0 (no, decreased in opportunities for physical activity before and after the earthquake; decreased walking speed; and no exercise habits). The independent variables were sex (male/female), over 75 years (yes/no), cohabitant (yes/no), temporary housing category (public/prefabricated/private sectors), current residence (owned/rented/public/public for disaster/hospitals and institutions/hospitals and institutions/other), and social support (yes/no). OR = odds ratio, 95% CI: 95% = confidence interval.

## Data Availability

The data presented in this study are available upon request from the corresponding author.

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
