# Peer review of "Factors Related to Physical Activity among Older Adults Who Relocated to a New Community after the Kumamoto Earthquake: A Study from the Viewpoint of Social Capital"

_ijerph, 2023, doi:10.3390/ijerph20053995_

Round 1
Reviewer 1 Report
1. I am not able to see a definite definition of social capital as per the study. kindly cite with appropriate references.
2. The need of the study shall become clearer with operational definitions of some terminologies
3. Objectives need to be more clearer in Abstract as well as in introduction.
4. Data analysis the why did the authors didn't used repeated measure design to check for interactions
5. Conclusion to be reframed
6. Figure describing the demographic distribution and data collection will make the presentation good.
Author Response
Thank you for your kind and constructive feedback on the article Factors Related to Physical Activity Among Older Adults who Relocated to a New Community After the Kumamoto Earth-quake: A Study from the Viewpoint of Social Capital. A Study from a Social Capital Perspective. We agree with your comments on the article and have revised the content as follows.

Reviewer 2 Report
The authors conducted an excellent study, with an original view on the practice of physical activity. The problem is well presented and constructed, as well as the methods. The results indicate that after the relocation of elderly people, especially women, to other locations due to the Kumamoto earthquake, they reduce their practice of physical activity. The discussion is also well presented, this being one of the few studies that did not verify the need for major or minor revision. Please see attachment:

Author Response

(The authors gave the same response as above.)

Reviewer 3 Report
Dear editors,
Thank you very much for having me to review this work.
Dear authors, thank you very much for the time you have dedicated to the elaboration of this paper which deals with such an important issue as physical activity in the elderly after an event that is a great stressor.
We would like to make a series of observations in order to contribute constructively to the work.
Firstly, it should be pointed out that the summary is well written, although it could include the variables analyzed to provide more information to future readers.
Regarding the introduction, they are based on a good review of the levels of physical activity in the elderly, but they leave aside the social factors that are the object of their study, as they point out in the table of contents. This is why these factors should be more present in the introduction.
Regarding the sample, they should indicate the % of men and women. Regarding the variables and the consultation made on them, the authors should indicate that although it is an ad hoc instrument, a review of similar questionnaires or in similar events has been carried out. Likewise, they should indicate whether it has undergone any validation process or statistical test for this purpose. The development and validation of the instrument is essential for the results to be credible.
The statistical tests proposed by the authors are good and meet their objectives.
In the results, the data on the characteristics of the sample should go before, as I have tried to highlight in the methodology section.
In the discussion they should try to compare results with other countries in similar situations but with other cultures, for example, Chile.
The conclusions are well stated and try to respond to the objectives.
Author Response

(The authors gave the same response as above.)

Reviewer 4 Report
Dear authors,
Thank you very much for this interesting article. However, in view of its publication, I think that a series of considerations should be taken into account:
ABSTRACT: I consider it very complete, but I would add the number of men and women and the average age in the same
KEYWORDS: try to make them different from those in the title
INTRODUCTION
Various studies are seen that speak of the reduction of activity after an earthquake. However, I would like to know if these people (or if there are studies) that previously analyzed the levels of physical activity (before the earthquakes occurred)
I consider it is interesting to be able to include some studies, for example of COVID (where many investigations were analyzed regarding the levels of physical activity that were carried out, or reasons for doing or not doing physical activity due to the situation).
I also find it very interesting to be able to see the coping capacity of people when faced with "big problems" like this type of face to look for possible solutions and, as you do in the study, see the most common barriers.
MATERIALS AND METHODS
Regarding the questionnaires, were validated questionnaires used? Was the Cronbach's alpha value of the questions analyzed? This is essential to ensure the validity of the questionnaires.
RESULTS
The first sentence would be included in the METHOD (sample selection criteria) “Of the collected questionnaires, 1,494 people were selected for analysis; These included adults aged 65 years and over who had relocated to a new community after the 162 earthquake.“
The tables are very clarifying, very well explained everything
DISCUSSION
I think that one of the most important aspects to consider is why so many people reduced their opportunities to engage in physical activity, but above all, what percentage of these people, "even though their opportunities were reduced" sought alternatives to continue doing physical activity.
Could it be that people switched from one form of physical activity to another? Did they just stop doing physical activity? I would like to see some recommendations for people who stop doing physical activity "because they no longer participate in community activities"
I consider in general it is a very good study, although it is advisable to include some aspects in the introduction about studies that describe "reasons for physical activity" that can be used when drawing conclusions, as a possible solution for those people who faced with a problematic phenomenon, they can look for alternatives to perform physical activity. For example, if "many people stop doing physical activity by not having collective sessions", being able to use digital media where everyone is connected, or videos from the Internet.
Very good work
Author Response

(The authors gave the same response as above.)

Round 2
Reviewer 1 Report
I feel a statistical check for the Test used should be undertaken by a professional statistician to check the appropriateness of the test used in the study
Author Response
Thank you for your kind and constructive feedback on the article Factors Related to Physical Activity Among Older Adults who Relocated to a New Community After the Kumamoto Earth-quake: A Study from the Viewpoint of Social Capital. We agree with your comments and have written a response to your comments in our response letter.
